# Insights into the Photoelectrocatalytic Behavior of gCN-Based Anode Materials Supported on Ni Foams

**DOI:** 10.3390/nano13061035

**Published:** 2023-03-13

**Authors:** Serge Benedoue, Mattia Benedet, Alberto Gasparotto, Nicolas Gauquelin, Andrey Orekhov, Johan Verbeeck, Roberta Seraglia, Gioele Pagot, Gian Andrea Rizzi, Vincenzo Balzano, Luca Gavioli, Vito Di Noto, Davide Barreca, Chiara Maccato

**Affiliations:** 1Department of Chemical Sciences, Padova University and INSTM, 35131 Padova, Italy; 2Laboratory of Applied Physical and Analytical Chemistry, Department of Inorganic Chemistry, Faculty of Science, University of Yaoundé, Yaoundé P.O. Box 812, Cameroon; 3CNR-ICMATE and INSTM, Department of Chemical Sciences, Padova University, 35131 Padova, Italy; 4EMAT and NANOlab Center of Excellence, University of Antwerp, 2020 Antwerpen, Belgium; 5Section of Chemistry for the Technology (ChemTech), Department of Industrial Engineering, University of Padova and INSTM, 35131 Padova, Italy; 6Interdisciplinary Laboratories for Advanced Materials Physics (i-LAMP), Dipartimento di Matematica e Fisica, Università Cattolica del Sacro Cuore, 25133 Brescia, Italy

**Keywords:** graphitic carbon nitride, CoPi, CoO, CoFe_2_O_4_, phthalates, wastewater remediation, oxygen evolution reaction

## Abstract

Graphitic carbon nitride (gCN) is a promising *n*-type semiconductor widely investigated for photo-assisted water splitting, but less studied for the (photo)electrochemical degradation of aqueous organic pollutants. In these fields, attractive perspectives for advancements are offered by a proper engineering of the material properties, e.g., by depositing gCN onto conductive and porous scaffolds, tailoring its nanoscale morphology, and functionalizing it with suitable cocatalysts. The present study reports on a simple and easily controllable synthesis of gCN flakes on Ni foam substrates by electrophoretic deposition (EPD), and on their eventual decoration with Co-based cocatalysts [CoO, CoFe_2_O_4_, cobalt phosphate (CoPi)] via radio frequency (RF)-sputtering or electrodeposition. After examining the influence of processing conditions on the material characteristics, the developed systems are comparatively investigated as (photo)anodes for water splitting and photoelectrocatalysts for the degradation of a recalcitrant water pollutant [potassium hydrogen phthalate (KHP)]. The obtained results highlight that while gCN decoration with Co-based cocatalysts boosts water splitting performances, bare gCN as such is more efficient in KHP abatement, due to the occurrence of a different reaction mechanism. The related insights, provided by a multi-technique characterization, may provide valuable guidelines for the implementation of active nanomaterials in environmental remediation and sustainable solar-to-chemical energy conversion.

## 1. Introduction

Over the past decades, the ever-increasing industrialization and world population growth have led to an unprecedented consumption of energy, mainly sustained by fossil fuels and, in parallel, to numerous harmful environmental effects, among which is water quality deterioration [1,2,3,4,5,6,7]. The latter is nowadays regarded as a global threat that is negatively impacting natural ecosystems [8] and is precluding large-scale access to clean water resources [9], whose safeguard is an urgent priority towards the wellbeing of society. In fact, the demand for both sustainable energy and drinking quality water is predicted to undergo a notable increase within 2050 [5,10]. Despite the fact that water is a largely abundant natural resource, the most Earth-abundant reservoir (~97% of the total H_2_O) is salt water from oceans or seas, whereas clean freshwater available for human consumption amounts to less than 1% [11], due to the daily discharge of several harmful compounds [1,2,3,6]. Unfortunately, many of these pollutants (including pesticides, pharmaceuticals, and dyes) are recalcitrant organic species that cannot be efficiently degraded by conventional treatments [1,6,11]. To this aim, a valuable alternative is offered by advanced oxidation processes, enabling efficient H_2_O purification under ambient conditions thanks to the generation of powerful oxidizing agents (i.e., hydroxyl radicals, •OH) [6,11], as in the Fenton process [10,11,12]. These kinds of photoactivated processes, driven by *n*-type active semiconductors such as TiO_2_ [12,13,14,15], are gaining a strategic importance for the simultaneous production of green H_2_ and water purification, which are of remarkable interest for real-world end-uses towards improved sustainability. As an alternative to TiO_2_-based photoanodes, gCN, a Vis-light active semiconductor, offers various concomitant advantages, including cost effectiveness, a negligible environmental impact, a favorable chemical reactivity, and good stability in various environments [16]. These features, coupled to the suitable band edge positions, have prompted its increasing investigation as an H_2_O splitting photoanode [4,5,17,18,19], though its use as a (photo)electrocatalyst for water decontamination has been less explored to date [20].

In the present work, we focused our attention on how relatively inefficient (photo)electrocatalysts for the oxygen evolution reaction (OER), the bottleneck of water splitting, can be, on the contrary, interesting platforms for •OH production in processes aimed at the degradation of aqueous pollutants. Specifically, we show how gCN flakes deposited by EPD on highly conductive and porous Ni foams can efficiently trigger OER if coupled with suitable Co-containing cocatalysts (CoO, CoFe_2_O_4_, or CoPi) [21,22]. The latter have been introduced in low amounts onto the pristine gCN by RF-sputtering or electrodeposition. A thorough characterization by complementary analytical tools evidenced that material OER performances were favorably boosted by the formation of suitable nano-junctions, suppressing detrimental electron-hole recombination and the favorable CoPi catalytic action [4,5,19,23]. Conversely, the use of bare gCN resulted in a more efficient production of •OH radicals and an enhanced degradation of aqueous KHP, a recalcitrant pollutant used as a standard benchmark [24,25] as confirmed by the “coumarin test” [14,26]. This aspect, i.e., the enhanced •OH radicals production over supported gCN flakes, which has not been addressed in detail up to date, is a key issue for the eventual mineralization of water soluble organic pollutants. In the following, after a multi-technique chemico-physical investigation of the developed electrocatalyst materials, their functional performances are presented and critically discussed as a function of the used synthesis and processing conditions. This work provides a new paradigm for constructing advanced photocatalysts for effective organic pollutant degradation and green energy conversion.

## 2. Materials and Methods

### 2.1. Material Preparation

In the present work, gCN-based systems were fabricated either in an Ar or air atmosphere (hereafter indicated as gCN^Ar^ or gCN^air^, respectively). Specifically, gCN powders (synthesized as described in the Appendix A) were finely grinded and subsequently used in EPD experiments. Accordingly, 40 mg of gCN and 10 mg of I_2_ were dispersed into a beaker containing 50 mL of acetone and the resulting mixture was sonicated for 20 min. Then, a pre-cleaned [27] Ni foam substrate (cathode) and carbon paper (anode), connected to a DC generator, were immersed in the resulting suspension at a distance of 10 mm. For bare and functionalized gCN^Ar^-based materials, experimental parameters for the carbon nitride deposition (EPD conditions: 5 min/50 V) were optimized according to our recently published work [19]. For gCN^air^-electrode materials, carbon nitride EPD conditions were preliminarily adjusted, testing different deposition times (from 1 to 15 min) and applied potentials (from 10 to 50 V), and selecting the ones (1 min/10 V) yielding the best photocurrents. Finally, the samples obtained from gCN^Ar^ and gCN^air^ powders were thermally treated in Ar (550 °C, 2.5 h) or in air (300 °C, 1 h), respectively.

The functionalization of gCN^Ar^-based samples with CoO or CoFe_2_O_4_ cocatalysts was performed by RF-sputtering from Co (Alfa Aesar^®^, Ward Hill, MA, USA; purity = 99.95%) or Co_3_O_4_-Fe_2_O_3_ targets (Neyco, Vanves, France; purity = 99.9%), using a custom-built, two-electrode plasmochemical reactor (ν = 13.56 MHz) and with the previously reported experimental settings [19]. The resulting materials were ultimately annealed in an Ar atmosphere at 500 °C for 2.5 h. Conversely, gCN^air^-based electrodes were functionalized with CoPi by electrodeposition. Experiments were carried out using an electrochemical workstation (Autolab PGSTAT-204 potentiostat/galvanostat, Utrecht, The Netherlands) and a three-electrode system composed of a Ni foam-supported material as the working electrode, a Pt coil as the counter-electrode, and an Ag/AgCl reference electrode. CoPi deposition was carried out from 0.5 mM CoCl_2_ aqueous solutions in 0.1 M potassium phosphate buffer (PBS, i.e., 0.1 M K_2_HPO_4_ and KH_2_PO_4_, pH = 7.1), following a previously reported procedure [23], until the highest OER current density was achieved (typically within 2–3 CV cycles). Appendix A reports the overall preparation procedure for gCN powders and the resulting Ni foam electrode materials, along with the most relevant synthesis conditions.

### 2.2. Material Characterization

#### 2.2.1. Characterization of gCN Powders

X-ray diffraction (XRD) measurements were performed using a Bruker (Karlsruhe, Germany) AXS D8 Advance Plus diffractometer, equipped a CuKα X-ray source (λ = 1.54051 Å). The average crystal size was estimated through the Scherrer equation. Analyses were carried out at the PanLab facility (Department of Chemical Sciences, Padova University) founded by the MIUR Dipartimento di Eccellenza grant “NExuS”. Fourier transform-infrared (FT-IR) spectra were recorded in transmittance mode on KBr pellets by means of a Thermo-Nicolet (Nicolet Instrument Corporation, WI, USA) Nexus 860 instrument (resolution = 4 cm^−1^). UV-Vis diffuse reflectance spectra were recorded on a Cary 5E (Varian, Palo Alto, CA, USA) spectrophotometer (spectral bandwidth = 1 nm), equipped with an integration sphere. The evaluation of band gap (*E*_G_) values was performed using the Tauc equation [f(R)*hν*]*^n^* vs. *hν*, where f(R) is the Kubelka–Munk function and R is the measured reflectance, assuming indirect and allowed transitions (*n = ½*) [18,28,29].

#### 2.2.2. Characterization of Electrode Materials

X-ray photoelectron spectroscopy (XPS) analyses were performed on a multiscan system (Omicron, ScientaOmicron, Uppsala, Sweden) using a Mg X-ray source (1253.6 eV) and a Phoibos 100 SPECS (SPECS, Berlin, Germany) analyzer. The binding energy values were corrected for charging phenomena by assigning a position of 284.8 eV to the adventitious C1s component [30]. After a Shirley-type background subtraction, curve fitting was carried out by the XPSpeak (Version 4.1) software [31], using Gaussian-Lorentzian sum functions. Atomic percentages (at.%) were calculated by peak area integration. Field emission scanning electron microscopy (FE-SEM) analyses were performed by collecting in-lens and backscattered electron signals using a Zeiss (Oberkochen, Germany) SUPRA 40VP instrument at primary beam acceleration voltages of 10–20 kV. Energy dispersive X-ray spectroscopy (EDXS) maps were acquired with an INCA x-act PentaFET Precision spectrometer. The analysis of electrode materials by transmission electron microscopy (TEM) required sample preparation using a ThermoFisher Scientific (Waltham, MA, USA) Helios Nanolab 650 dual-beam focused ion beam instrument. A Pt protection layer was deposited in order to avoid material damage, and the obtained sample foils were thinned to a thickness < 50 nm. Electron diffraction, EDXS, high-resolution scanning TEM (HRSTEM), electron energy loss spectroscopy (EELS), and image simulation were employed for a detailed characterization. The electron diffraction and EDXS analyses were performed on a ThermoFisher Scientific Osiris microscope equipped with a Super-X windowless EDX detector system, operated at 200 kV. High-resolution EELS spectra were acquired using a state-of-the-art double-corrected and monochromated ThermoFisher Scientific Titan 80-300 microscope operated at 120 kV. Further details are reported in the Appendix A (Appendix A).

### 2.3. Functional Tests

OER electrochemical tests were carried out both in the dark and under irradiation, using an Autolab three-electrode workstation (PGSTAT204 potentiostat/galvanostat), with the prepared samples and a Pt coil serving as a working electrode and counter-electrode. An Hg/HgO electrode, typically employed in alkaline media, was selected as the reference. The electrolyte was a 0.1 M KOH (pH = 12.9) aqueous solution. The working electrode was exposed to a white light LED source (intensity ≈ 150 mW/cm^2^). For pollutant degradation, the cell was filled with 50 mL of aqueous solution containing 1 mM KHP and 0.1 M KOH. Then, Ni foam-supported samples were tested as (photo)anodes under constant stirring at a fixed bias voltage of 1.5 V vs. the reversible hydrogen electrode (RHE). Hereafter, all potentials are expressed with respect to RHE. At regular time intervals, aliquots of the KHP solution were collected and the residual pollutant content was measured by flow injection analysis combined with electrospray mass spectrometry (FIA-ESI/MS) [32]. For each sample, FIA-ESI/MS measurements were carried out three times, revealing a <3% deviation among repeated experiments. FIA-ESI/MS analyses were performed with a LCQFleet ion trap instrument (ThermoFisher Scientific), operating in negative ion mode, coupled with a Surveyor LC Pump Plus (ThermoFisher Scientific). The entrance capillary temperature, ion source temperature, and voltage were set to 275 °C, 300 °C, and 4 kV, respectively. The initial KHP solutions were diluted 10 times with H_2_O/acetonitrile (50/50, *v*/*v*) with 0.1% of formic acid. The resulting solutions (25 µL) were injected into the mass spectrometer, using a mixture of H_2_O/acetonitrile (80/20, *v*/*v*) with 0.1% formic acid as eluent (flow rate = 0.3 µL/min). To ensure reproducibility and reliability of the obtained results, each degradation experiment was repeated twice.

The “coumarin test” was used to probe •OH radical production by monitoring the formation of photoluminescent 7-hydroxy-coumarin [14,26,33]. Although such method does not allow for an absolute quantitation of •OH radicals, it is a highly sensitive and valuable tool for the determination of relative activities between different photocatalysts [33]. To this aim, electrochemical experiments were carried out under illumination on a 1 mM coumarin solution in 0.1 M PBS, using the above-described cell and experimental conditions. Fluorescence spectra were collected in the 342–700 nm wavelength range using an FLS 1000 fluorimeter (Edinburgh Instruments, Livingston, United Kingdom). The following settings were used: excitation wavelength/bandwidth = 332/1.5 nm; emission bandwidth = 2.5 nm; optical path = 1 cm. Measurements were also carried out on Ni foam-supported materials using the same instrument, with an excitation wavelength of 410 nm.

## 3. Results and Discussion

As already mentioned, gCN powders were synthesized by the thermal condensation of melamine at 550 °C for 4 h either in Ar (gCN^Ar^) or in air (gCN^air^), to investigate the influence of processing conditions on the properties and performances of the resulting Ni foam-supported samples.

The XRD patterns of the gCN powders synthesized in Ar and air are compared in Figure 1a. In both cases, the reflections at ≈13.2° and ≈27.3° were attributed to the packing of tri-s-triazine units in the (100) crystallographic plane, and to the interplanar (002) stacking of gCN sheets, respectively [17,34,35]. The broad and weak peaks at ≈44° and ≈57° can be ascribed to gCN (300) and (004) planes [35]. The corresponding mean nanocrystal sizes were directly affected by the used reaction atmosphere (Figure 1a, inset).

The FT-IR spectra of gCN^Ar^ and gCN^air^ powders (Figure 1b) displayed similar features, with a multi-component band in the region of 1200–1650 cm^−1^ due to typical stretching modes of aromatic heptazine (C_6_N_7_) heterocycles, along with the characteristic heptazine breathing mode at 808 cm^−1^ [17,34,36]. The broad band in the 2900–3500 cm^−1^ range resulted from stretching vibrations of the –OH groups at the edges of carbon nitride sheets [36,37] and of uncondensed –NH_x_ groups (x = 1, 2), whose presence was also responsible for the N–H deformation mode at 885 cm^−1^ [17,18,19,36]. The presence of such moieties was confirmed by XPS data (see below). The analysis of gCN powders by UV-Vis diffuse reflectance spectroscopy evidenced a relatively steep edge at λ ≈ 500 nm (Appendix A), due to interband electronic transitions [38,39]. Tauc plot analyses yielded *E*_G_ values of ≈2.6 eV for both gCN^air^ and gCN^Ar^ (Appendix A) [17,19].

Both kinds of powders were subsequently used as precursors for EPD on the Ni foams of gCN systems, which were eventually functionalized with CoO and CoFe_2_O_4_ (via RF-sputtering), or CoPi (via electrodeposition). Figure 2 reports representative SEM images for samples gCN^air^ and gCN^air^–CoPi, along with EDXS elemental maps for the latter sample. As can be observed, µm-sized gCN particles with a flake-like morphology were uniformly dispersed over the Ni foam, yet leaving uncovered some substrate regions. For gCN^air^–CoPi, EDXS maps for carbon and nitrogen highlighted the discontinuous deposit nature. Nonetheless, the O and P signals overlapped with the gCN ones, thus indicating that CoPi deposition preferentially took place over carbon nitride grains rather than on the uncovered Ni foam regions. Although cobalt could not be revealed by EDXS, its presence was clearly confirmed by XPS analyses (see below and the Appendix A).

As far as gCN^Ar^-containing samples are concerned, a lower Ni foam coverage was revealed (Appendix A). In the case of functionalized specimens, the presence of ultra-dispersed CoO or CoFe_2_O_4_ nanoparticles (samples gCN^Ar^–CoO and gCN^Ar^–CoFe_2_O_4_, respectively) was clearly revealed by high resolution TEM analyses. More specifically, the former specimen contained spherical aggregates with a size smaller than 10 nm. The *d*-spacings and high angle annular dark-field high resolution scanning TEM (HAADF-HRSTEM) results (Figure 3a) were in line with the occurrence of cubic CoO. This conclusion was confirmed not only by the matching between experimental and simulated HRSTEM images (Figure 3a), but even by the electronic structure analysis of the target nanoparticles. EELS measurements revealed indeed a good agreement between the experimental Co L_2,3_ (Figure 3b) and O K (Appendix A) edges and those of a CoO reference, thus allowing for the exclusion of Co_3_O_4_ presence. For specimen gCN^Ar^–CoFe_2_O_4_, TEM analysis on the observed nanoparticles highlighted the formation of spinel-type CoFe_2_O_4_ aggregates (Figure 3c). Even in this case, a good agreement was observed between experimental and simulated HRSTEM images, and the formation of phase-pure CoFe_2_O_4_ was confirmed by the EELS spectra of Co L_2,3_, Fe L_2,3_, and O K edges (Figure 3d and Appendix A).

The surface chemical composition and elemental chemical states were investigated by XPS. Survey spectra (Appendix A) displayed the presence of C and N photopeaks and, for gCN^Ar^–CoO and gCN^Ar^–CoFe_2_O_4_ specimens, even the presence of Co or Co and Fe signals. The occurrence of Ni peaks indicated an incomplete coverage of the underlying Ni foam. The quantitative analyses (see data in the caption for Appendix A) suggested the occurrence of nitrogen-deficient systems. As observed in Figure 4a, Appendix A, the C1s photopeak could be fitted by three components (see also Figure 4b): ***a***, due to adventitious contamination (BE = 284.8 eV) [40,41]; ***b***, related to C atoms in uncondensed C–NH_x_ (x = 1, 2) groups on carbon nitride ring edges (BE = 286.2 eV) [42,43]; ***c***, due to C in N-C=N moieties (BE = 288.2 eV) [28,40,43,44]. The weight of component ***b*** underwent an appreciable increase upon going from bare gCN^Ar^, to gCN^Ar^–CoO, up to gCN^Ar^–CoFe_2_O_4_ (Figure 4c). These data suggested an enhancement of defects related to the presence of amino groups, resulting from gCN^Ar^ bombardment during RF-sputtering processes, according to the order gCN^Ar^ < gCN^Ar^–CoO < gCN^Ar^–CoFe_2_O_4_. Recent reports have demonstrated that the presence of surface defects in electrode materials can effectively tune the electronic structure and charge density, and thus improve the redox reactivity [45,46]. In particular, the above-mentioned N defects can beneficially suppress charge carriers recombination, boosting the ultimate material photoactivity [42,47] (see also below). These conclusions were in line with the N1s peak fitting results (see Appendix A and related comments).

For both gCN^Ar^–CoO and gCN^Ar^–CoFe_2_O_4_ samples, the Co2p photopeak (Figure 4d) presented well-developed shake-up structures on the high BE side of the main components, indicating the occurrence of Co(II) centers [28,30,48]. In fact, for gCN–CoO, the BE positions [BE(Co2p_3/2_) = 781.1 eV; spin-orbit separation (SOS) = 15.7 eV] were in line with previously reported data for CoO [28,29]. In the case of gCN^Ar^–CoFe_2_O_4_, Co2p_3/2_ BE was slightly higher (781.3 eV), in agreement with the data pertaining to CoFe_2_O_4_ systems [36,49]. The formation of the latter oxide, confirmed by TEM and related analyses (see above), was also supported by the Fe2p peak features (Figure 4e; BE(Fe2p_3/2_) = 710.6 eV; SOS = 13.3 eV) [36,49,50]. In the case of gCN^air^–CoPi, the Co2p spectral features were in accordance with the literature data for CoPi systems (Appendix A) [51]. O1s spectra are reported in the Appendix A (see Appendix A and pertaining observations).

Figure 5a,b display linear sweep voltammetry scans collected on the target specimens, where the current increase at potentials > 1.5 V vs. RHE corresponds to the OER. As can be observed, for Ar-treated samples (Figure 5a) the difference between light and dark currents was relatively small (0.1–0.4 mA/cm^2^ at E > 1.6 V). This result was related to a high and uneven carbon nitride loading onto Ni foams, negatively affecting the system conductivity and producing a modest light response. This phenomenon, in turn, could be ascribed to the potential delay caused by the resistance–capacitance couple in series, particularly evident at high scan rates [52]. In a different way, gCN^air^ and gCN^air^–CoPi, containing a lower gCN amount (Figure 5b), yielded much higher photocurrent densities (1.0–2.5 mA/cm^2^ at E > 1.6 V), due to a more uniform gCN flakes distribution.

It is worth highlighting that both Ar- and air-treated samples revealed a remarkable current density improvement upon gCN functionalization, with either CoO, CoFe_2_O_4_, or CoPi. The anodic peaks between 1.35 and 1.5 V correspond to the MO → MO(OH) reaction usually occurring on the surface of Ni and Co oxide nanoparticles [27]. The higher peak intensity for NiF^Ar^ and gCN^Ar^ in comparison to the homologous gCN^air^ and NiF^air^ samples could be attributed to an enhanced Ni oxidation, related, in turn, to a less uniform substrate coverage by gCN in the case of Ar-treated samples. The resulting systems were tested as (photo)electrocatalysts for the purification of aqueous solutions containing KHP, a recalcitrant pollutant, whose decomposition is obtained in the presence of •OH radicals [54]. Preliminary degradation tests were carried out to assess the electrochemical performances of gCN^Ar^–CoO and gCN^Ar^–CoFe_2_O_4_ both in the dark and under irradiation (Appendix A). As can be observed, the specimens yielded a ~50% decrease in the KHP concentration after 1 h, while longer times did not result in further significant variations. Such a result suggests that the decrease in KHP concentration vs. time was mainly due to KHP adsorption on gCN flakes via π–π interactions [55]. A slight additional decrease for longer times, especially under illumination, was observed for gCN^Ar^–CoFe_2_O_4_. This result was also in line with indications provided by the “coumarin test” (see below, Figure 7a), revealing that only a modest •OH production took place in the case of gCN^Ar^–CoO, whereas for gCN^Ar^–CoFe_2_O_4_, •OH formation was enhanced.

Based on the higher gCN^air^ and gCN^air^–CoPi photoresponses (Figure 5), such specimens were subsequently tested in light-activated KHP degradation, analyzing the process kinetics by FIA-ESI/MS. Whereas the sample gCN^air^ promoted a significant and continuous decrease in the KHP concentration as a function of time, gCN^air^–CoPi was almost completely inactive (Figure 6a). Interestingly, the electrochemical performances of gCN^air^ towards KHP degradation were preserved upon repeating the same experiment after three months, highlighting a good material stability. The experimental trend in Figure 6b, in agreement with a pseudo-first order kinetics [56], could be fitted with either a single or a double exponential function. The obtainment of a better match with experimental results in the latter case (see the corresponding χ^2^ value in Figure 6b) indicates that two distinct processes, i.e., adsorption and decomposition, take place simultaneously on the material surface.

The different behaviors of gCN^air^ and gCN^air^–CoPi, apparently unexpected since an opposite activity trend was detected in the OER (Figure 5), can be explained considering that organic pollutant degradation is directly dependent on •OH production and does not involve the four-electron mechanism of OER. In alkaline media, the OER process involves in fact the adsorption of hydroxyl ions on the catalyst active sites. Subsequently, hydroxyl ions are oxidized to hydroxyl radicals that are hence deprotonated and oxidized again, yielding surface-adsorbed oxygens. If two of such species are close enough, they can finally react together to produce an O_2_ molecule [57]. In a different way, KHP degradation is mostly dependent on the anodic production of •OH radicals, whose formation takes place on the catalyst surface when electrogenerated holes react with adsorbed H_2_O/OH^−^ species [2,57]. The attack of •OH radicals to KHP typically involves the initial hydroxylation of the aromatic ring and/or decarboxylation of its side chains with the formation of several hydroxybenzoic acids and benzene diols. Then, the aromatic ring is opened and the formation of several low molecular weight carboxylic acids takes place, followed by further oxidation steps, finally yielding to CO_2_ and H_2_O [54,58,59].

To attain a deeper insight into the above processes, photoluminescence experiments with coumarin solutions were carried out to probe the formation of the luminescent 7-hydroxy-coumarin, occurring in the presence of hydroxyl radicals [14,26]. Such tests clearly revealed a significant production of •OH radicals only for the sample gCN^air^, whereas a negligible photoluminescence was detected for gCN^air^–CoPi (Figure 7a), in good agreement with the relative KHP degradation trends of Figure 6a. How should this result be interpreted?

As reported by Ehrmaier et al. [60], the production of •OH radicals on the gCN surface under illumination occurs on heptazine ring units. Heptazine photoexcitation to the lowest π–π* singlet yields the formation of an electron-hole pair (Figure 7b). The generated hole can subsequently oxidize a water molecule attached to heptazine via a hydrogen bond. The subsequent electron transfer from water to heptazine is followed by the release of the hydrogen-bonded proton, resulting in the final formation of heptazinyl and •OH radicals. This mechanism can experience appreciable variations in the presence of surface trap states for holes (Figure 7b, lower part), as Co(II) centers in the Co-containing cocatalysts, whose occurrence appreciably reduces the effective hydroxyl radical content. We believe that photogenerated holes are trapped on Co centers so that the mechanism occurring on CoPi nanoparticles is activated, while the oxidation of OH^−^ ions on heptazine units is depressed. In fact, solid-state luminescence spectra collected on the samples gCN^air^ and gCN^air^–CoPi (Figure 8) revealed that CoPi introduction completely modified the charge transport dynamics on the gCN surface, producing a strong luminescence quenching attributable to Co(II) trap states.

## 4. Conclusions

In this study, gCN flakes were grown on Ni foam substrates by EPD under different conditions, and eventually functionalized with low amounts of highly dispersed CoO, CoFe_2_O_4_, or CoPi cocatalysts. Preliminary water splitting experiments revealed that OER performances of the developed systems were improved by gCN functionalization with the above Co-containing species. In a different way, when the same materials were tested as photoelectrocatalysts for KHP degradation, the most active system turned out to be bare gCN, free from any cocatalyst. This apparently surprising behavior was traced back to the different reaction mechanisms involved in water splitting or KHP degradation. In particular, for bare gCN as such, the enhanced generation of •OH radicals (very limited, on the contrary, for systems functionalized with Co-based cocatalysts) efficiently triggered the target pollutant degradation, as demonstrated by means of complementary characterization studies.

The applicative potential of the present work is highlighted by the fact that these electrodes can be easily prepared even with much larger dimensions, and can be easily recovered for subsequent recycling at variance with powdered gCN photocatalysts. In perspective, the outcomes of our research activities yield a promising prospect for the design and engineering of active nanomaterials featuring tailored properties for solar-to-chemical energy conversion and environmental remediation.

## Figures and Tables

**Figure 1 nanomaterials-13-01035-f001:**
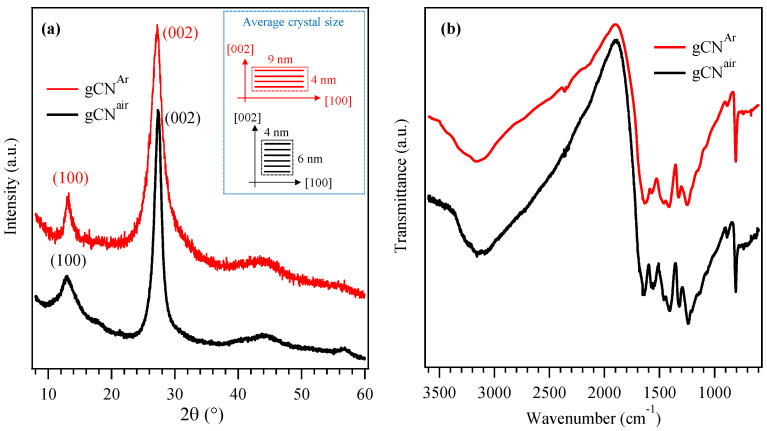
(**a**) XRD patterns of gCN^Ar^ and gCN^air^ powders. The average gCN crystallite sizes along the [100] and [002] directions are reported in the inset. (**b**) Corresponding FT-IR spectra.

**Figure 2 nanomaterials-13-01035-f002:**
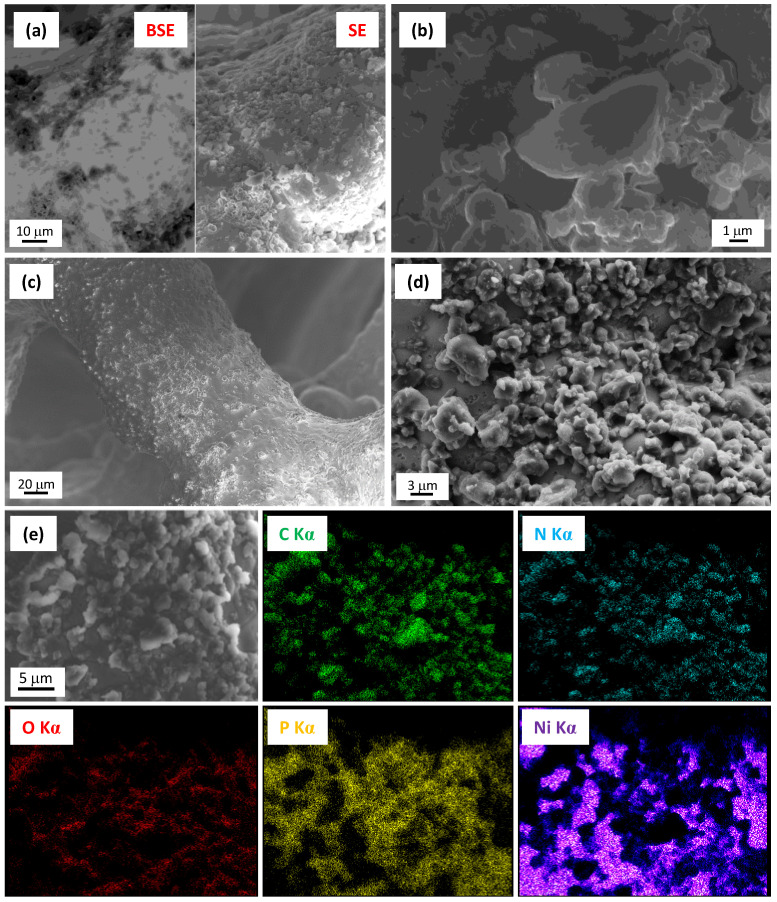
FE-SEM micrographs for samples (**a**,**b**) gCN^air^ and (**c**,**d**) gCN^air^–CoPi. In panel (**a**), left and right images were obtained collecting back-scattered (BSE) and secondary (SE) electrons, respectively. The gCN presence corresponds to dark regions in (**a**). EDXS elemental maps for specimen gCN^air^–CoPi recorded on the electron image in (**e**) are also reported.

**Figure 3 nanomaterials-13-01035-f003:**
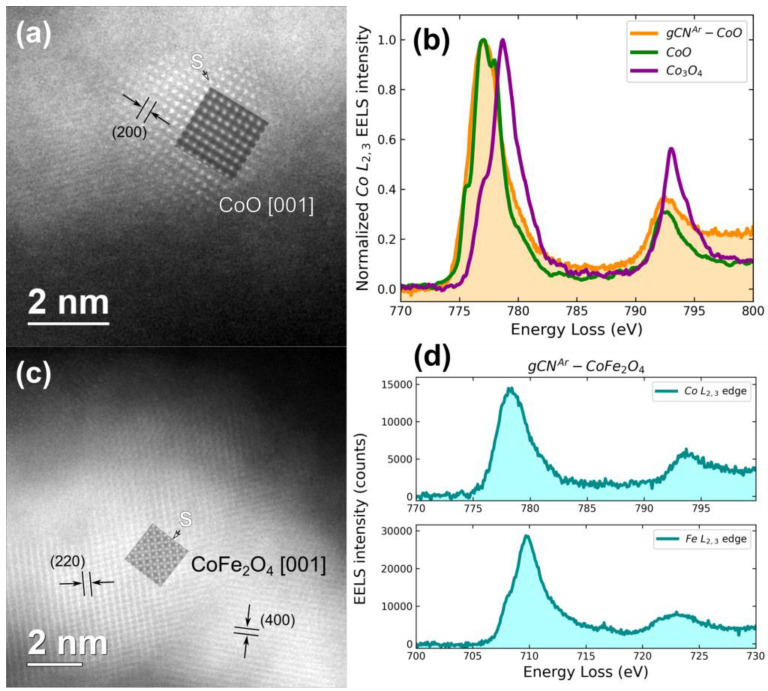
HAADF-HRSTEM images (left) and EELS spectra (right) of the Co L_2,3_ and eventually Fe L_2,3_ edges acquired on Co-containing nanoparticles for samples gCN^Ar^–CoO (**a**,**b**) and gCN^Ar^–CoFe_2_O_4_ (**c**,**d**). ‘S’ marks the simulated HRSTEM images.

**Figure 4 nanomaterials-13-01035-f004:**
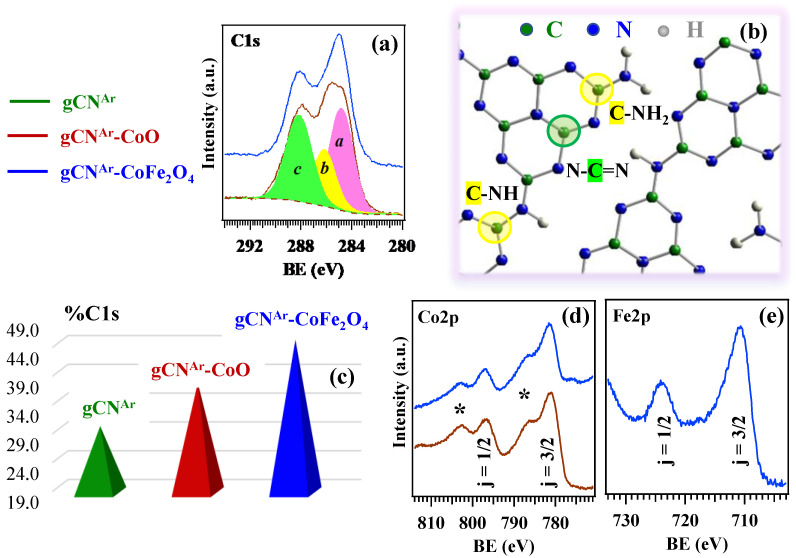
XPS characterization of gCN^Ar^ deposits on Ni foams before and after functionalization with CoO and CoFe_2_O_4_ via RF-sputtering. (**a**) C1s photoelectron peaks for gCN^Ar^–CoO and gCN^Ar^–CoFe_2_O_4_. (**b**) Sketch of gCN structure [53], in which non-equivalent C sites are marked. Color codes as in panel (**a**). (**c**) Percentage contribution of the ***b*** component to the overall C1s photopeak for the target specimens. Calculation was performed excluding the adventitious component ***a***. (**d**) Co2p and (**e**) Fe2p signals for gCN^Ar^–CoO and gCN^Ar^–CoFe_2_O_4_. In (**d**), stars (*) indicate shake-up peaks.

**Figure 5 nanomaterials-13-01035-f005:**
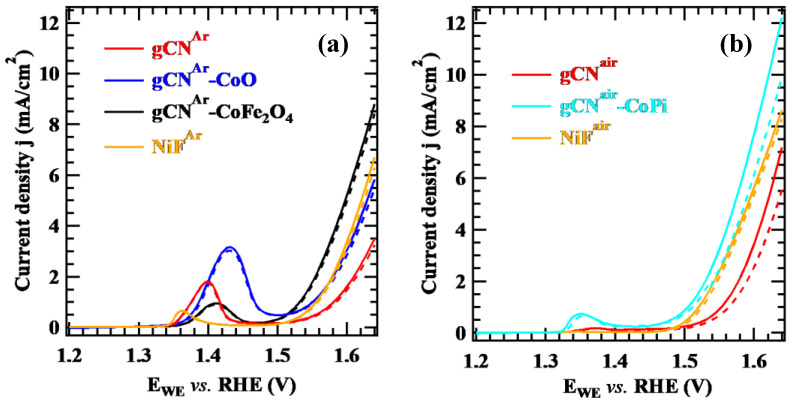
Linear sweep voltammetry scans collected in 0.1 M KOH solution on (**a**) Ar-treated and (**b**) air-treated supported samples. Continuous and dotted curves have been collected under irradiation and in the dark, respectively. For each sample set, bare Ni foams (NiF) scans are also reported.

**Figure 6 nanomaterials-13-01035-f006:**
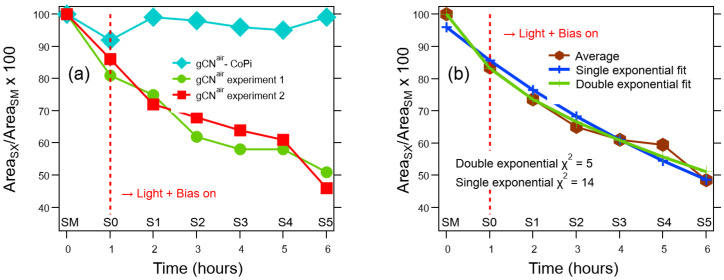
(**a**) KHP degradation tests carried out on gCN^air^ and gCN^air^–CoPi. KHP residual content was determined from the *m/z* = 165 hydrogen phthalate peak area as follows: Area_SX_/Area_SM_ × 100, where SM = mother solution; S0 = solution after adsorption/desorption equilibrium in the absence of external bias; SX = solution after testing each specimen under irradiation for X hours (X = 1÷5). For sample gCN^air^, experiment 2 (red curve) was performed three months after experiment 1 (green plot). (**b**) Average of the two plots reported in (**a**) for sample gCN^air^ and fit of the corresponding experimental results both with a single or double exponential function.

**Figure 7 nanomaterials-13-01035-f007:**
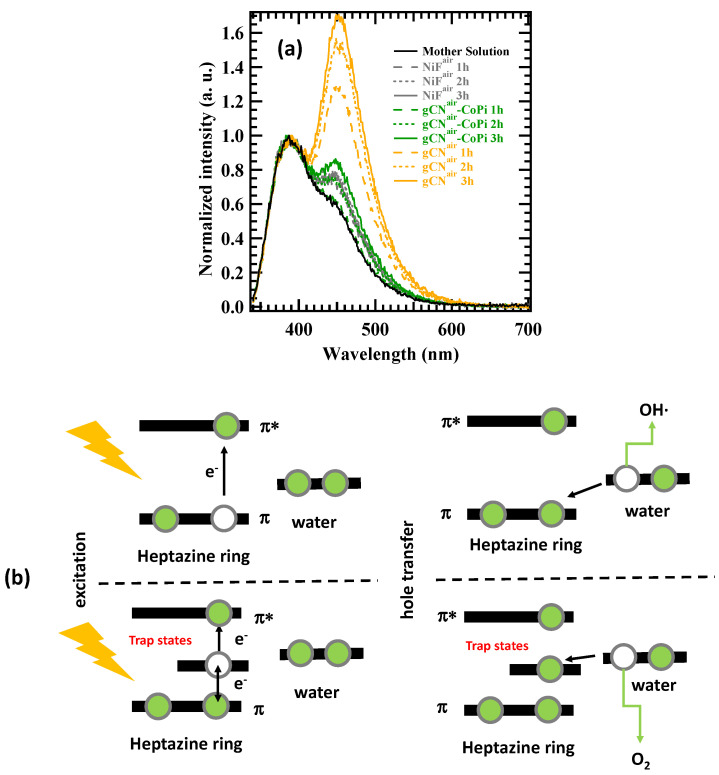
(**a**) Luminescence spectra obtained from 1 mM coumarin solution in 0.1 M phosphate buffer after 1, 2, or 3 h of photoelectrochemical work under illumination. Curves corresponding to bare Ni foam are also displayed. The luminescence peak at 390 nm is due to coumarin emission [14]. The increased emission at ~450 nm for gCN^air^ indicates an appreciably higher production of 7-hydroxy-coumarin for this specimen. (**b**) Sketch of the mechanism for the production of •OH radicals on gCN surface [60]. The upper and lower parts of the image sketch are processes taking place in the absence and in the presence of trap states [Co(II) centers], respectively.

**Figure 8 nanomaterials-13-01035-f008:**
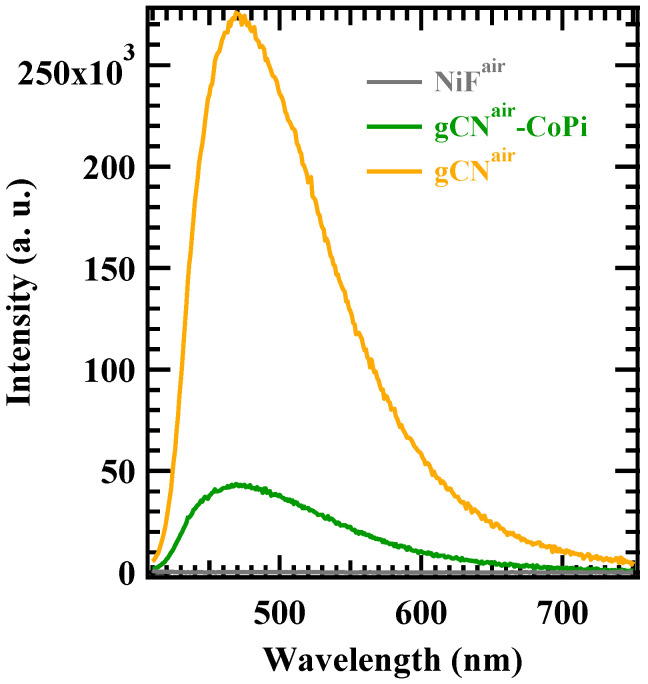
Solid-state luminescence spectra collected on samples gCN^air^ and gCN^air^–CoPi. The spectrum for the bare Ni foam is also reported for comparison.

## Data Availability

Data supporting this study are available within the article.

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
