# Peer review of "Insights into the Photoelectrocatalytic Behavior of gCN-Based Anode Materials Supported on Ni Foams"

_nanomaterials, 2023, doi:10.3390/nano13061035_

Round 1

Reviewer 1 Report

This manuscript reported a facile synthesis of gCN flakes on Ni foam substrates by electrophoretic deposition (EPD), followed by functionalization with different Co-based cocatalysts (CoO, CoFe2O4, cobalt phosphate (CoPi), by RF-sputtering, which were employed as anode materials for photoelectrocatalytic applications.

Some specific suggestions are given as follows:

 (1) In terms of material preparation, what is the effect of gCN prepared by thermal treatment in Ar and air on further functionalization? What are the differences?

(2) The authors stated “Although cobalt content could not be revealed by EDXS, its presence was clearly confirmed by XPS analyses (see below).” on page 5, lines 225 and 226, how to explain. The content of Co-based cocatalyst can be given.

(3) The authors mention that the presence of OH radicals plays a decisive role in (photo)electrocatalytic degradation of KHP. Therefore, in this work, a test for trapping OH radicals by EPR spectroscopy analysis should be provided.

(4) The mechanism of photoelectrocatalytic decomposition of water and degradation of KHP needs to be fully explained.

(5) In the introduction section, some recently published papers about this work regarding materials and applicatyions should be cited such as Chemical Engineering Journal 453 (2023) 139875, Journal of Materials Chemistry A 2020, 8, 24053-24064, and Small, 2022, 18(5), 2104507.

(6) In Part 3, there is only subheading 3.1. The authors should check and improve the manuscript profoundly.

Author Response

Reviewer 1

Comment: This manuscript reported a facile synthesis of gCN flakes on Ni foam substrates by electrophoretic deposition (EPD), followed by functionalization with different Co-based cocatalysts (CoO, CoFe2O4, cobalt phosphate (CoPi), by RF-sputtering, which were employed as anode materials for photoelectrocatalytic applications.

Some specific suggestions are given as follows:

Reply: We thank the Reviewer for the detailed and critical examination of our paper. His/her comments stimulated us not only to improve the manuscript, but also to consider in detail the potential of our work and its broader significance. Our answers to his/her specific remarks are reported below.

Comment: (1) In terms of material preparation, what is the effect of gCN prepared by thermal treatment in Ar and air on further functionalization? What are the differences?

Reply: Although characterization results revealed some influence of the thermal treatment in Ar vs. air on the microstructural features of gCNAr and gCNair powders (see nanocrystal dimensions in Fig. 1a), the reaction atmosphere had a modest effect on the functional properties of the corresponding Ni-foam supported samples (containing only carbon nitride) that featured a similar activity in the production of •OH radicals. Conversely, gCN decoration with CoO, CoFe2O4, CoPi exerted a strong detrimental effect on •OH radical production/KHP degradation, due to the presence of Co(II) centers acting as surface traps for holes. The fact that, for samples gCN-CoO and gCN-CoFe2O4, an appreciable decrease of KHP amount vs. time was revealed in Fig. S10 should be mainly attributed to KHP adsorption and only partially to its degradation via ∙OH radical production. In order to better convey this message to the reader, the main paper text has been modified on page 9, lines 314-317.

Comment: (2) The authors stated “Although cobalt content could not be revealed by EDXS, its presence was clearly confirmed by XPS analyses (see below).” on page 5, lines 225 and 226, how to explain. The content of Co-based cocatalyst can be given.

Reply: As requested by the Reviewer, Co atomic percentages, estimated by XPS, have been now provided in the Supporting Material section (see caption for Figures S5 and S7), and recalled in the main paper text on page 5, line 225.

Comment: (3) The authors mention that the presence of OH radicals plays a decisive role in (photo)electrocatalytic degradation of KHP. Therefore, in this work, a test for trapping OH radicals by EPR spectroscopy analysis should be provided.

Reply: We agree with the Reviewer that EPR experiments using radical traps like the TEMPO molecule might provide very useful indications concerning the production of •OH radicals. Nonetheless, we believe that tests with coumarin are reliable as well, taking into account that the luminescence signal of 7-hydroxy-coumarin is very intense, making this method very sensitive and allowing thus to probe even extremely low concentrations of hydroxyl radicals. In this regard, it is important to highlight that we also experimentally observed a very good agreement between the 7-hydroxy-coumarine luminescence signal and the degree of KHP decomposition, as now better highlighted in the main paper text on page 10, lines 360-361. Additionally, to further support the validity of coumarine tests we have now quoted in the manuscript (page 4, lines 180-182) a novel work by Žerjav et al. (see Reference 33) where authors state: “Photoluminescence technique using probe compounds to determine the amount of generated •OH radicals can be an effective approach to compare the photocatalytic activity of different catalysts worldwide, although this is certainly not the only available benchmarking tool...” and “The results show that exact quantitation of generated •OH radicals through these two probes is prevented by many drawbacks, mainly by nonspecific and nonstoichiometric reactions with •OH radicals and low stability of measured reaction products (7- hydroxy-coumarin). However, a determination of relative •OH radical formation rates between different photocatalysts is potentially possible through measurements of monohydroxylated products of coumarin”.

Comment: (4) The mechanism of photoelectrocatalytic decomposition of water and degradation of KHP needs to be fully explained.

Reply: In order to satisfy the Reviewer request, the mechanisms of water decomposition and KHP degradation in the presence of hydroxyl radicals have now been described on page 9, lines 343-349 and page 10, lines 350-355, also addressing the reader to additional literature papers on such topics.

Comment: (5) In the introduction section, some recently published papers about this work regarding materials and applications should be cited such as Chemical Engineering Journal 453 (2023) 139875, Journal of Materials Chemistry A 2020, 8, 24053-24064, and Small, 2022, 18(5), 2104507.

Reply: The suggested papers have been cited in the manuscript (now Refs. 44, 45 and 46; see also Ref. 20 in the Supporting Material).

Comment: (6) In Part 3, there is only subheading 3.1. The authors should check and improve the manuscript profoundly.

Reply: We apologize for our oversight, and have now removed subheading 3.1. In addition, as suggested, the whole manuscript text has been revised line-by-line with utmost care, in order to improve its clarity, quality and impact.

Reviewer 2 Report

Title: Insights into the photoelectrocatalytic behavior of gCN-based anode materials supported on Ni foams

This comparative study of bare g-CN with its modified versions towards water splitting and KHP degradation is interesting. Besides this, there are still some concerns which need to be addressed before its acceptance for publishable. I hope that the mentioned points will be resolved well as much as possible. Please see the comments below thoroughly.

Recommendation: Revision

Major Comments:

1. In the abstract (line no. 26-30), it is mentioned the limited use of bare g-CN due to its lack of active site and fast charge recombination, while it is claimed (line no. 36-39)that bare g-CN was more effective towards KHP degradation. So, its better to specify the limiting use of bare g-CN in a sense for particular field. Reader will feel vague after reading this abstract if its left unarranged in a sense of objective of the study. 2. I could not find the characterizations (XRD and FTIR) for functionalized/modified samples; g-CN-CuO and g-CN-CuFe2O4. The metallic phase from XRD and FTIR can strengthen the other characterizations of your materials.  3. Did the authors optimize the conditions for experimental parameters?

Minor comments:

1. Correct the Figure 1 (a) line heading 2 theta NOT 2teta. 2. Author has jumped from section 3 (Results and Discussion) to section 5 (conclusion). Is there missing of section 4 or miswritten of section 5 instead of section 4? 3. Are there additional subsections in section 3? If not, then no need to mention subsection 3.1.  4. To insight/elaborate of the degradation phenomenon/techniques/mechanisms from various material designations as well as technologies, you can prefer the following literatures as follows.https://doi.org/10.1016/j.cej.2021.129312    

Chemosphere, Volume 309, Part 1, December 2022, 136638;http://dx.doi.org/10.1016/j.chemosphere.2022.136638

Catalysts 202010(5), 546; https://doi.org/10.3390/catal10050546

Author Response

Reviewer 2

Comment: This comparative study of bare g-CN with its modified versions towards water splitting and KHP degradation is interesting. Besides this, there are still some concerns which need to be addressed before its acceptance for publishable. I hope that the mentioned points will be resolved well as much as possible. Please see the comments below thoroughly.

Reply: We are grateful to the Reviewer for the positive evaluation of our paper, and we acknowledge his/her observations. As explained in the following, we have done our best to revise the paper accordingly.

Major Comments:

Comment: 1. In the abstract (line no. 26-30), it is mentioned the limited use of bare g-CN due to its lack of active site and fast charge recombination, while it is claimed (line no. 36-39) that bare g-CN was more effective towards KHP degradation. So, its better to specify the limiting use of bare g-CN in a sense for particular field. Reader will feel vague after reading this abstract if its left unarranged in a sense of objective of the study.

Reply: For sake of clarity, the Abstract has been amended according to the Reviewer suggestion.

Comment: 2. I could not find the characterizations (XRD and FTIR) for functionalized/modified samples; g-CN-CoO and g-CN-CoFe2O4. The metallic phase from XRD and FTIR can strengthen the other characterizations of your materials. 

Reply: We thank the Reviewer for the pertinent comment. The analysis of Ni foam-supported samples by XRD was preliminarily carried out by us, both in Bragg-Brentano and glancing incidence configuration, but the obtained patterns were noisy and analytically not useful. This result was related to the corrugation/porosity of the substrate (hindering an optimal sample alignment, also due to the in-depth dispersion of the deposited material into the Ni foam channels), as well as to the strong fluorescence from the Ni foam itself. In a similar way, the analysis of the developed electrode materials by FTIR was not technically feasible in transmittance mode as the substrate is not sufficiently transparent, nor in ATR (attenuated total reflectance), due the fragility of the Ge crystal in the ATR accessory.

Comment: 3. Did the authors optimize the conditions for experimental parameters?

Reply: For bare and functionalized gCNAr-based materials, experimental parameters for carbon nitride, CoO and CoFe2O4 deposition were optimized according to our recently published work (now Ref. 19). For gCNair-electrode materials, carbon nitride EPD conditions were preliminarily optimized testing different deposition times (from 1 to 15 min) and applied potentials (from 10 to 50 V), and selecting the ones (1 min/10 V) yielding the best photocurrents. In the case of CoPi electrodeposition, experimental conditions were optimized following the original report by Nocera et al. (now Ref. 23), and using 2 to 3 CV cycles, until the highest OER current was reached. These details have now been better highlighted in the Materials and Methods section, that has been significantly restructured (see page 3, lines 100-118).

Minor comments:

Comment: 1. Correct the Figure 1 (a) line heading 2 theta NOT 2teta.

Reply: Fig. 1 line heading has been modified.

Comment: 2. Author has jumped from section 3 (Results and Discussion) to section 5 (conclusion). Is there missing of section 4 or miswritten of section 5 instead of section 4?

Reply: We apologize for the oversight (there was no section 4 in the original submission and the Conclusions were numbered as section 5 by mistake) and have now corrected the mistake.

Comment: 3. Are there additional subsections in section 3? If not, then no need to mention subsection 3.1. 

Reply: As already indicated in the answer to comment 6 from Reviewer 1, we have removed the mention to subsection 3.1.

Comment: 4. To insight/elaborate of the degradation phenomenon/techniques/mechanisms from various material designations as well as technologies, you can prefer the following literatures as follows. https://doi.org/10.1016/j.cej.2021.129312 Chemosphere, Volume 309, Part 1, December 2022, 136638; http://dx.doi.org/10.1016/j.chemosphere.2022.136638 Catalysts 2020, 10(5), 546; https://doi.org/10.3390/catal10050546.

Reply: The suggested works have been cited in the revised paper (now Refs. 7, 8, and 15).

Reviewer 3 Report

This article describes the study of synthesized gCN powders with and without Co-containing cocatalysts and immobilized them on Ni foam substrates by EPD. The results showed improved OER performance with Co-containing gCN for water splitting, but bare gCN was more effective for KHP degradation. The study highlights the applicative potential of the developed electrodes, which can be easily prepared and recovered for use in wastewater treatment and hydrogen generation. However, some issues must be solved before it is considered for publication. If the following problems are well-addressed, this reviewer believes that the essential contribution of this article is vital for photocatalytic water purification.

1.      How does the functionalization of gCN with different metal oxides (CoO, CoFe2O4, CoPi) affect its electrochemical performance for the purification of aqueous solutions containing KHP? Can the differences in electrochemical performance between gCNair and gCNair-CoPi be attributed to differences in the mechanism of hydroxyl radical production?

2.      What is the effect of surface trap states for holes, such as Co(II) centers, on the production of hydroxyl radicals during the degradation of aqueous solutions containing KHP?

3.      How do the differences in charge transport dynamics on the gCN surface, induced by the introduction of CoPi, affect the production of hydroxyl radicals and the overall electrochemical performance of gCN for the purification of aqueous solutions containing KHP?

4.      Can the stability of the electrochemical performances of gCN towards KHP degradation over time be improved by modifying its substrate coverage or by using different metal oxide functionalization? What is the long-term stability of the developed anode materials under different operating conditions?

5.      Can the different reaction mechanisms observed for water splitting and KHP degradation be reconciled or optimized for dual applications? Can the developed anode materials be integrated into practical wastewater treatment systems, and what are the design considerations and challenges for such applications?

Author Response

Reviewer 3: 

Comment: This article describes the study of synthesized gCN powders with and without Co-containing cocatalysts and immobilized them on Ni foam substrates by EPD. The results showed improved OER performance with Co-containing gCN for water splitting, but bare gCN was more effective for KHP degradation. The study highlights the applicative potential of the developed electrodes, which can be easily prepared and recovered for use in wastewater treatment and hydrogen generation. However, some issues must be solved before it is considered for publication. If the following problems are well-addressed, this reviewer believes that the essential contribution of this article is vital for photocatalytic water purification.

Reply: We are grateful to the Reviewer for the very positive evaluation of our work. We have carefully taken into account the raised issues and modified the manuscript accordingly, as described in the following.

Comment: 1. How does the functionalization of gCN with different metal oxides (CoO, CoFe2O4, CoPi) affect its electrochemical performance for the purification of aqueous solutions containing KHP? Can the differences in electrochemical performance between gCNair and gCNair-CoPi be attributed to differences in the mechanism of hydroxyl radical production?

Reply: As already explained in the answer to comment 1 from Reviewer 1, although characterization results revealed some influence of the thermal treatment in Ar vs. air on the microstructural features of gCNAr and gCNair powders (see nanocrystal dimensions in Fig. 1a), the reaction atmosphere had a limited effect on the functional properties of the corresponding Ni-foam supported samples (containing only carbon nitride) that featured a similar activity in the production of •OH radicals. Conversely, gCN decoration with CoO, CoFe2O4, CoPi exerted a strong detrimental effect on •OH radical production/KHP degradation, due to the presence of Co(II) centers acting as surface traps for holes. The fact that, for samples gCN-CoO and gCN-CoFe2O4, an appreciable decrease of KHP amount vs. time was observed (Fig. S10), should be mainly attributed to KHP adsorption and only partially to its degradation via •OH radical production. In order to better convey this message to the reader, the main paper text has been modified on page 9, lines 314-317.

Comment: 2. What is the effect of surface trap states for holes, such as Co(II) centers, on the production of hydroxyl radicals during the degradation of aqueous solutions containing KHP?

Reply: We acknowledge the Referee observation. The effect of trap states is to drive the OER reaction on Co-containing particles and not on gCN heptazine rings. In any case, the mechanisms of water decomposition and KHP degradation in the presence of hydroxyl radicals have now been better described on page 9, lines 343-349 and page 10, lines 350-355, also addressing the reader to additional literature papers on such topics.

Comment: 3. How do the differences in charge transport dynamics on the gCN surface, induced by the introduction of CoPi, affect the production of hydroxyl radicals and the overall electrochemical performance of gCN for the purification of aqueous solutions containing KHP?

Reply: We believe that photogenerated holes are trapped on Co centers so that the mechanism occurring on CoPi nanoparticles is activated, while the oxidation of OH- ions on heptazine units is depressed. For sake of clarity, these observations have been reported on page 11, lines 378-381.

Comment: 4. Can the stability of the electrochemical performances of gCN towards KHP degradation over time be improved by modifying its substrate coverage or by using different metal oxide functionalization? What is the long-term stability of the developed anode materials under different operating conditions?

Reply: The gCN sample supported on Ni foams worked for about 10 hours in 0.1 M KOH solution (the decomposition was repeated twice), yielding very similar results. The amount of gCN was optimized to obtain the highest photocurrent and, at the same time, to avoid detachment from the electrode usually occurring for higher carbon nitride loadings. We have never noticed any appreciable electrode degradation. In the present work, the stability of gCNair-CoPi towards KHP degradation was not assessed through repeated measurements since this sample was found to be inactive already during the first degradation experiment (see Fig. 6a). Conversely, as also stated in the answer to comment 4 from Reviewer 4, all other samples have been tested twice to check for stability, as now clarified on page 4, lines 177-178.

Comment: 5. Can the different reaction mechanisms observed for water splitting and KHP degradation be reconciled or optimized for dual applications? Can the developed anode materials be integrated into practical wastewater treatment systems, and what are the design considerations and challenges for such applications?

Reply: We believe that our findings can provide useful guidelines to properly optimize material properties for water splitting or for KHP degradation. If the target aim is water purification, the best candidate is bare gCN, free from any functionalizing agent. In a different way, in view of water splitting applications, gCN decoration with Co-containing cocatalysts offers better performances and an improved stability (see Ref. 19). These issues have also been better addressed by restructuring the Conclusions section (page 11), as also indicated in the comment 1 from Reviewer 4.

Reviewer 4 Report

Journal: nanomaterials

Manuscript number: nanomaterials -2257165

Title: Insights into the photoelectrocatalytic behavior of gCN-based anode materials supported on Ni foams

The present study reports on a facile synthesis of gCN flakes on Ni foam substrates by electrophoretic deposition (EPD) and on the subsequent system decoration with different Co-based cocatalysts (CoO, CoFe2O4, cobalt phosphate (CoPi) in low amounts, by radio frequency sputtering (RF-sputtering) or electrodeposition. The paper is interesting although some issues should be addressed before publication. Major revision is suggested to further improve its quality. Specific suggestions are provided below.

1.        What is the biggest flash point in this article should be provided in the introduction?

2.        There are too many abbreviations in the article.

3.        Did the author do a parallel experiment?

4.        How about the stability of gCNair-CoPi? (Applied Catalysis B: Environmental 2020. 273: 119051 and Separation and Purification Technology 2023, 312: 123412 may help and recommended to be cited)

Author Response

Reviewer 4

Comment: The present study reports on a facile synthesis of gCN flakes on Ni foam substrates by electrophoretic deposition (EPD) and on the subsequent system decoration with different Co-based cocatalysts (CoO, CoFe2O4, cobalt phosphate (CoPi) in low amounts, by radio frequency sputtering (RF-sputtering) or electrodeposition. The paper is interesting although some issues should be addressed before publication. Major revision is suggested to further improve its quality. Specific suggestions are provided below.

Reply: We thank the Reviewer for the positive evaluation of our work. We have carefully taken into account all the raised issues and suggestions for improvement, as described in the following, and the paper has been carefully revised accordingly.

Comment: 1. What is the biggest flash point in this article should be provided in the introduction?

Reply: We acknowledge the Referee observations. Accordingly, we have made significant and explicit modifications in the "front matter" of the paper (Abstract, Keywords, Introduction, Conclusions), in order to better evidence the manuscript originality and impact with respect to the state-of-the-art in the field.

Comment: 2. There are too many abbreviations in the article.

Reply: Following the Referee suggestions, we have amended the manuscript so as to keep the number of abbreviations to the minimum possible.

Comment: 3. Did the author do a parallel experiment?

Reply: As far as KHP degradation is concerned, FIA-ESI/MS measurements were repeated three times on each specimen, revealing a <3% deviation among parallel experiments. This piece of information has now been added on page 4, lines 169-170.

Comment: 4. How about the stability of gCNair-CoPi? (Applied Catalysis B: Environmental 2020. 273: 119051 and Separation and Purification Technology 2023, 312: 123412 may help and recommended to be cited).

Reply: We acknowledge the Reviewer comment concerning the importance of material stability, and have quoted the two suggested references (now Refs. 9 and 16). In the present work, the stability of gCNair-CoPi towards KHP degradation was not assessed through repeated measurements because this sample was found to be inactive already during the first degradation experiment (see Fig. 6a). Conversely, all other samples have been tested twice to check for stability, as now clarified on page 4, lines 177-178.

Round 2

Reviewer 2 Report

Accept